# Training History, Cardiac Autonomic Recovery from Submaximal Exercise and Associated Performance in Recreational Runners

**DOI:** 10.3390/ijerph19169797

**Published:** 2022-08-09

**Authors:** Matic Špenko, Ivana Potočnik, Ian Edwards, Nejka Potočnik

**Affiliations:** 1Medical Faculty, Institute of Physiology, University of Ljubljana, 1000 Ljubljana, Slovenia; 2Centre for Cardiovascular and Metabolic Neuroscience, Department of Neuroscience, Physiology & Pharmacology, University College London, London WC1E 6BT, UK

**Keywords:** training recovery, heart rate variability, heart rate recovery, endurance exercise

## Abstract

This study investigated the effect of prolonged exertion on cardiac parasympathetic (cPS) reorganization and associated aerobic performance in response to repeated short-lasting submaximal exercise bouts (SSE) performed for 7 days following prolonged exertion. In 19 recreational runners, heart rate (HR) and HR variability (HRV) indices (lnRMSSD, lnHF, and lnLF/HF) were monitored pre- and post-submaximal graded cycling performed on consecutive days following a half-marathon (HM) and compared with the baseline, pre-HM values. Additionally, HR recovery (HRR), aerobic performance, and rate of perceived exertion (RPE) were determined. HR, HRV indices, and HRR were tested for correlation with exercise performance. A significant time effect was found in HR, HRR, and HRV indices as well as in aerobic performance and RPE during the study period. Most of the measured parameters differed from their baseline values only on the same day following HM. However, HRR and HR measured in recovery after SSE were additionally affected one day following the half-marathon yet in opposite directions to those recorded on the same day as the HM. Thus, postSSE HR and HRR exhibited a bivariate time response (postSSE HR: 102 ± 14 bpm; *p* < 0.001; 82 ± 11 bpm; *p* = 0.007 vs. 88 ± 11 bpm; HRR in 30 s after SSE cessation: 14.9 ± 4.9 bpm; *p* < 0.001; 30.1 ± 13.3 bpm; *p* = 0.006 vs. 24.4 ± 10.8 bpm), potentially indicating a cPS dysfunction phase on the same day and cPS rebound phase one day following HM reflected also in consecutive changes in aerobic power. Correlations were found between the changes in measured cardiac indices with respect to baseline and the changes in aerobic performance indices throughout the study period. The effect of exercise history on cPS reorganization is more pronounced in response to SSE than at rest. Accordingly, we conclude that SSE performed repeatedly on a daily basis following prolonged exertion offers a noninvasive tool to evaluate the impact of training history on cPS recovery and associated aerobic power output in recreational athletes.

## 1. Introduction

There are many exercise characteristics affecting the time course of post-exercise recovery: exercise modality [1], duration, intensity [2], muscle mass involved [3] and fitness level [4] which were studied thoroughly in the last decade. However, a pre-exercise training history as a key determinant of post-exercise recovery has been overlooked. As reported in many studies [5,6], complete recovery may take several days following exercise. Therefore, when exercising on a daily basis, it is very likely that a successive exercise bout will be performed before the recovery from the previous one is completed. The term ‘training recovery’ was introduced by Bishop and colleagues as a recovery between successive work-outs, and they concluded in their review that the occurrence of overtraining is the simultaneous product of both the recovery and the work-out [7]. While recovery affects exercise performance and vice versa, the planning of training sessions regarding the exercise/recovery balance is fundamentally important: it may lead either to athlete’s exhaustion with attenuated performance or overcompensation with improved performance [8]. Accordingly, understanding of the cumulative effects of multiple successive training sessions and their relationship to recovery period is required [9] to predict the current performance of the athlete as well as to avoid overreaching and overtraining.

The autonomic nervous system (ANS) plays a pivotal role in the regulation of the cardiovascular system (CVS) to ensure optimal function during both exercise and recovery. Dynamic exercise is associated with decreased parasympathetic and increased sympathetic nerve activity to the heart [10] while both processes are reversed during recovery [3]. Accordingly, during multiple successive exercise bouts and their recovery, a complex interplay between sympathetic and parasympathetic activity to the heart establishes a dynamic state, which may either lead to better CVS adaptation to exercise or provoke its compromised response, reflecting in improved or impaired athlete’s performance. Hence, an in-depth knowledge of day-to-day cardiac ANS activity dynamics may be necessary to promote optimal healthy performance, prevent sudden cardiac events related to exercise, and potentially recognize functional overreaching.

In a recent review, Van Hooren and Peake [11] reported that the practical relevance of findings about recovery after a single exercise bout is limited regarding training recovery. The same authors analysed dozens of articles studying the exercise performance during training recovery after exhausting practice or competition yet none of them explored the time course of aerobic capacity and corresponding cardiac ANS response to recurrent aerobic exercise bouts following exhaustive exercise. To date, we have not identified studies that have addressed this issue either in elite or in recreational athletes. On the other hand, Stanley and his colleagues [9] presented a review article addressing heart rate variability (HRV) changes in response to a single but not repetitive exercise. There are plenty of reports about long-term changes in cardiac ANS activity (cANA) throughout exhaustive training programs lasting several weeks [12]. There is little to no data on the short-term cANA response to successive submaximal short-lasting aerobic exercise bouts following exhaustive exercise on daily bases; nevertheless, this is a frequently used training regime in recreational athletes. A need for additional data on post-exercise vagal-related HRV indices is outlined by Manresa-Rocamora, to enable the tracking of cardiac parasympathetic activity (cPS) changes in relation to functional overreaching and overtraining [13].

The measurement of HRV is considered a convenient non-invasive assessment tool for monitoring cANA, especially its vagal component. Furthermore, HRV is a promising tool in the area of sport for the purpose of exercise load, training adaptation, and fitness level evaluation [1]; however, there are no studies addressing the balance between recurrent submaximal exercise following prolonged exhaustive exercise by means of HRV. As proposed by Plews and his colleagues, measurement of resting and post-exercise HRV is a reliable method to monitor individual adaptation to training [14].

Upon exercise cessation, time dependent recovery of heart rate (HR) and HRV toward pre-exercise values was demonstrated by many authors [3,5,9]. Rapid recovery occurs in the initial minutes after exercise [15] but it may take more than 48 h for the activity of the autonomic nervous system to be completely restored [3,5,14]. Peçanha [16] differentiated between vagal reactivation, responsible for HR drop in the first minutes after exercise, and vagal modulation which occurs with a certain time delay. A highly reproducible measure of vagal reactivation is heart rate recovery (HRR) whereas HRV reflects vagal modulation [17]. Pre-exercise and post-exercise HRV-derived parasympathetic indices as well as HRR, although all vagally mediated, have been suggested to represent independent aspects of cPS [18] and respond differentially to endurance exercise [19].

To date, we have not identified any studies that have addressed how a prolonged exhaustive exercise (PE) affects the cardiovascular response to successive short-term submaximal dynamic exercise (SSE) bouts performed repeatedly throughout recovery after PE. Therefore, our study was designed to simulate day-to-day training regime consisting of PE in the form of a half-marathon run followed by graded submaximal cycling as a model of SSE performed successively during post-PE recovery. We aimed to assess preSSE and postSSE cANA as well as associated performance and the rate of perceived exertion in post-PE recovery, from 1 h up to 1 week. We have minimized the effect that different exercise intensities and modalities may have on recovery process by conducting the same SSE protocol over the whole study and the effect of gender, age, and individual aerobic fitness by employing a repeated measures study design, where the participants themselves serves as their own controls.

Our study focused on evaluating the response of cardiac parasympathetic autonomic activity to recurrent submaximal exercise performed following prolonged exertion. We aimed to determine how different pre-exercise cANA provoked by half-marathon as a model of prolonged exertion influences the physiological response to short-lasting submaximal exercise, associated power output, and RPE during a one week follow-up. For this purpose, HR and HR variability were measured at rest and after submaximal graded cycling performed before, on the same day, and 24 h, 48 h, and 1 week following a half-marathon run in recreational runners. Heart rate recovery was determined for each submaximal exercise session. We also aimed to test if any correlations exist between the changes in HR, HRR, or HRV indices and aerobic performance. It was hypothesized that prolonged exertion influences cardiac autonomic response to submaximal exercise in a time-dependent manner and may potentially indicate cardiac vulnerability, increased cardiac function, or early signs of overtraining. We also hypothesized that changes in HR, HRR, or HRV indices regarding exercise history may be used for monitoring exercise performance.

## 2. Materials and Methods

### 2.1. Subjects

A total of 20 recreational runners, training regularly more than 3 times a week, were recruited to participate voluntarily in this repeated measures study, designed in accordance with the Declaration of Helsinki and approved by the Ethics Committee of Republic Slovenia (Date 19 October 2021/No. 0120-126/2021/10). One participant dropped out after the half-marathon run because of an injury; all others (N = 19, 13 men), aged 40.4 ± 15.2, with body mass index 23.0 ± 2.7 kg/m^2^, finished the whole study period. All the subjects provided written informed consent before participation. Their physical examination and history revealed no autonomic dysfunction, chronic diseases, medication usage, or smoking. Their ECG and arterial blood pressure were normal. Individual maximal heart rate (HRmax) was determined by $HRmax=205.8-\left(0.685\;age\right)$ [20], proposed by Inbar and coworkers [21], recommended as the most accurate for incremental exercise testing [22]; age should be given in years, returning the value of HRmax in bpm.

### 2.2. Procedures

The study [21] was carried out in a climate-controlled laboratory between 2 and 6 pm. The subjects refrained from physical exertion for at least 3 days before the first exercise test and were asked not to perform additional physical activities during the experimental period. Subjects were not allowed to consume any alcohol, caffeine, or tobacco for at least 2 h before the beginning of each exercise test and were asked to eat a light meal 1 h before coming to the laboratory. Each participant visited the laboratory 4 times in one week for 5 measurements. During the first visit, PE in the form of 21 km run was performed on the predefined outside track. The run distance was covered on a relatively flat route, close to the laboratory on partly sunny days with temperatures ranging from 17 °C to 20 °C and humidity between 55% and 65%. Participants were asked to run as fast as they can to complete the half-marathon and were encouraged during the run. They were equipped with HR monitoring devices (Polar V800, Polar Electro, Kempele, Finland) to trace their HR when running. Participants were allowed to drink ad libitum during the run and were provided with body-tempered water for hydration after completing the run to minimize the effect of water drinking on cardiac vagal tone [23]. They did not ingest significant amounts of food when running but did eat snacks during the first 20 min after running cessation (1 g/kg of carbohydrate) according to the nutrition guidelines for recovery after a marathon run (Nutrition and Athletic Performance).

SSE was repeated at 5 different time points: before (before), one hour (day0), 24 h (day1), 48 h (day2), and 8 days (day7) after PE as shown on Figure 1A. Values recorded during SSE in the ‘before’ session are referred to as baseline values. SSE consisted of 5 min rest in a sitting position followed by a graded exercise test on the Ergoselect 100 (Ergoline, Bitz, Germany) cyclo-ergometer, starting at 40 W for 3 min and continuing with workload increase in steps of 50 W every 3 min until target heart rate (HRpeak, Istanbul, Turkey), defined as 85 % of individual Hrmax, was attained (Figure 1B). Immediately afterwards, the subjects stopped exercising and remained still in the seated position for 15 min to passively recover from SSE. Throughout SSE, the pedal frequency was kept at 60 revolutions per minute and the subjects breathed spontaneously.

Before each SSE, the subjects were weighed and whole-body resistance and reactance were obtained by bio impedance analysis (BIA) (BIA-101, Akern, Firenze, Italy) using foot-to-hand measurement to assess hydration [24]. After that, the participants rested for 30 min sitting in the laboratory when equipped with standard ECG lead II electrodes and a thoracic band to record the chest movements for breathing frequency determination. During SSE, ECG (Finometer model 2 (Amsterdam, The Netherlands) and respiratory movements (respiratory belt TN1132/ST, AdInstruments, Sydney, Australia) were recorded simultaneously at 500 Hz using WinDaq data acquisition software (DataQInstruments Inc., Akron, OH, USA, ZDA).

### 2.3. Heart Rate and Heart Rate Variability Analysis

RR interval duration (time between two successive R waves in the ECG) was determined beat by beat from ECG using aHRV_file_preparation software (Nevrokard, Slovenia) where artefacts and premature beats were corrected. During the SSE protocol, RR intervals from two time periods of 180 s were selected for HRV analysis: the last three minutes of sitting rest (preSSE) and from the 12th to 15th minute after cessation of cycling (postSSE) (Figure 1B). HR and root mean square of successive differences (RMSSD) as a time domain marker of parasympathetic modulation were computed from RR interval time series by aHRV_analysis_software (Nevrokard); natural logarithm of RMSSD (lnRMSSD) was also calculated. The following frequency domain parameters were determined by the autoregressive method: the power of high-frequency band (HF; 0.15–0.40 Hz) and the LF/HF ratio, where LF means the power of low-frequency band (0.04–0.15 Hz) [25]. HF and LF/HF were expressed in natural logarithmic scale (lnHF). HF power was used as a marker of parasympathetic modulation, whilst the LF/HF ratio was used as a marker of sympathovagal balance.

Heart rate recovery in 30 (HRR30) and 60 s (HRR60) were defined as the differences between HRpeak and HR recorded at these time points following exercise and were recorded for each subject during each SSE session.

### 2.4. Indices of Aerobic Performance

Maximal power output (Pmax) is an important determinant for the aerobic endurance performance and can be easily determined from the HR versus power relationship in the steady state [26]. As there is a linear dependence of power output on HR during steady state cycling, maximal power output can be determined indirectly by extrapolation of the linear regression line to HRmax even if graded cycling was ended at submaximal level [27]. To obtain HR versus power linear relationship coefficient (HR/Pslope) for each participant, the subject’s mean steady state HR was determined at each power step during SSE and linear regression between both variables was performed by IBM SPSS Statistics, version 27. For each SSE, Pmax of participants was determined by the indirect method [26]. Additionally, peak power output (Ppeak), i.e., power output at which target HR was achieved during graded cycling and SSE discontinued was recorded. Ppeak could serve as an approximate measure of Pmax.

As a direct measure of aerobic performance, a maximal steady state HR (HRstmax), determined as an average over the last minute of cycling at the subject’s last stationary power output, was introduced. Last stationary power output was defined as the power output that a particular subject can handle before HR drift was observed during graded cycling on day0. For a particular subject, the same submaximal power output was considered over all five SSEs.

### 2.5. Subjective Markers

The subjective perception of exertion during SSE (RPE) was assessed immediately after cycling cessation on the Borg scale [28], ranging from 6 (very, very easy) to 20 (maximal stress). Borg chose this range of values to reflect the heart rate of healthy individuals doing physical work; the Borg score multiplied by 10 can be used to compare the subjective perception of exertion with the heart rate during the work in question.

The subjects rated their perception of physical recovery based on leg soreness using a visual analogue scale (VAS), which was reported as a reliable marker of delayed onset muscle soreness [6]. The VAS consisted of a 100 mm line, the endpoints of which were labelled as ‘no pain’ (left) and ‘unbearable pain’ (right). Before each SSE, the subjects were asked to point their general amount of muscle pain in the scale.

### 2.6. Other Parameters

Bioelectrical impedance analysis [24] was applied to predict the total body water (TBW) and extracellular water (ECW) volumes in the participants before SSE at all five time points.

Breathing frequency (BF) was determined by WinDaq analysis software (DataQInstruments Inc., Akron, OH, USA, ZDA) based on the rhythmic thoracic belt movements for both above mentioned time intervals (preSSE and postSSE). All BFs were checked to fall inside the HF band of frequency domain HRV.

### 2.7. Statistical Analysis

The study was performed according to a repeated measures design, and a level of confidence *p* < 0.05 was selected. Statistical analysis was completed using IBM SPSS Statistics, version 27 (IBM, New York, NY, USA). Data were tested for normality and log transformed if not normally distributed. We compared mean differences in measured parameters over time (before, day0, day1, day2 and day7) preSSE and postSSE with a repeated measures one-way ANOVA (rANOVA). The assumption of sphericity was checked by Mauchlly’s test; Greenhouse–Geisser or Huynh–Feldt corrections were applied when sphericity assumption was violated as published elsewhere [29]. A significant main effect will be reported in the form: F (effect degrees of freedom, error degrees of freedom) = *F* value, p = *p* value, partial eta squared (η^2^) = *η*^2^ value. Variables, measured or determined only post-SSE (e.g., HRR, HR/Pslope, Ppeak, Pmax, HRstmax, and RPE) or preSSE (VAS) over time (before, day0, day1, day2, and day7) were also compared by rANOVA when normally distributed. When detecting a significant time effect, corresponding contrast tests were used to identify differences between means on day0, day1, day2, day7, and corresponding baseline (before PE), as well as between preSSE and postSSE if applicable. For post hoc comparisons, a least significant difference test was applied, and Bonferroni correction was used to eliminate type I errors in multiple comparisons. The VAS and Ppeak distributions were not normal even in logarithmic scale, so Friedman rANOVA on Ranks was performed for these data. When a main effect was detected, Wilcoxon signed-rank test was applied to identify the differences. In addition, a magnitude-based inferences method was applied for practical significance. Effect sizes (ES, 95% confidence interval) in RPE, HR/Pslope, Pmax, and HRstmax were calculated using the pooled standard deviation [30] at the day0, day1, day2, and day7 with respect to baseline (before PE). Cohen’s thresholds for small, moderate, and large standardized differences in means were set at ESs 0.2, 0.5, and 0.8, respectively [30].

Pearson’s product–moment correlation coefficients (r) were used to estimate the bivariate correlations between individual cardiac autonomic activity indices (HR pre- and post-SSE, lnRMSSD pre- and post-SSE, lnHF pre- and post-SSE, HRR30, and HRR60) and subsequent SSE performance indices (Pmax, HRstmax, RPE) for each SSE separately and combined over the whole experimental trial. To minimize the effect of diversity in age and sex of our participants, the individual change of a particular parameter was defined as a ratio between its value at any time point after PE (day0, day1, day2, day7) and baseline value. These individual changes, denoted by the suffix % (HR preSSE%, HR postSSE%, lnRMSSD preSSE%, lnRMSSD postSSE%, lnHF preSSE%, lnHF postSSE%, HRR30%, HRR60%) were tested for correlations to the subsequent performance indices change (Pmax%, HRstmax%, and RPE%, respectively). The following criteria were adopted to interpret the magnitude of correlations: r ≤ 0.1—trivial; r = 0.1–0.3—small; r = 0.3–0.5—moderate; r = 0.5–0.7—large; r > 0.7—very large [31].

## 3. Results

### 3.1. Subject Characteristics

Baseline characteristics of the subjects, their baseline regular training schedule, the score of the half-marathon run completed as a part of the study protocol, and mean HR achieved during this run are shown in Table 1.

### 3.2. HR-Derived Measures

There was a significant time effect found in HR (F(4.527; 81.486) = 69.51; *p* < 0.001; η^2^ = 0.79), HRR30 (F(4; 72) = 19.54; *p* < 0.001; η^2^ = 0.52), and HRR60 (F(4; 72) = 22.26; *p* < 0.001; η^2^ = 0.55).

Figure 2 shows mean ± SD of HR preSSE and postSSE (Figure 2A) as well as HRR30 and 60 (Figure 2B) at all five time points: before and following (day0, day1, day2, day7) PE. The *p*-values given refer to significant changes in the time course of the variables compared to baseline, pre-PE values. HR post-SSE, HRR30, and HRR60 exhibited a clear and significant biphasic response on the day0 and day1 compared to baseline (Figure 2A,B). Additionally, significantly lower HR post-SSE compared to baseline was demonstrated on the day2 (Figure 2A). No significant biphasic response was revealed in HR preSSE. As can be seen from Figure 2A, HR in the recovery after SSE was significantly higher at all five time points compared to simultaneous preSSE values (*p* < 0.001).

Time course of the change in HRR30 with respect to its baseline value following the half-marathon is illustrated in Figure 3B. On day1, 2, and 7, HRR30% exceeded the baseline value and is significantly increased with respect to day0 (*p* values indicated in Figure 3B).

### 3.3. HRV-Derived Measures

There was a significant time effect found in lnRMSSD (F(3.08; 52.52) = 22.5035; *p* < 0.001; η^2^ = 0.57), lnHF (F(9; 162) = 6.81; *p* < 0.001; η^2^ = 0.28) and ln(LF/HF) (F(4.93; 83.80) = 6.59; *p* < 0.001; η^2^ = 0.28).

As can be seen from Figure 2E, lnRMSSD post-SSE was significantly lower at all five measuring time points compared to preSSE values (*p* < 0.001). On day0, both pre- and postSSE lnRMSSD decreased significantly compared to baseline. Concerning frequency domain HRV indices, both, lnHF (Figure 2D) and ln(LF/HF) (Figure 2C) differed significantly comparing preSSE and postSSE, except on day 0 when they remained unchanged (*p* = 0.648 and 0.118, respectively). On the day0, resting lnHF decreased but ln(LF/HF) increased significantly compared to baseline, whereas neither lnHF nor ln(LF/HF) post-SSE revealed a meaningful change over time (Figure 2C,D).

Time course of changes in lnRMSSD and lnHF with respect to corresponding baseline values following the half-marathon are illustrated in Figure 3A,C. On day1, 2, and 7, lnRMSSD and lnHF overshot the baseline value yet were more pronounced post-SSE than preSSE (Figure 3A,C). Changes in lnRMSSD preSSE and postSSE as well as lnHF preSSE showed a significant increase on day1, 2, and 7 compared to day0, whereas no significant changes were observed concerning lnHF post-SSE (Figure 3A,C).

### 3.4. Indices of Aerobic Performance

There was a significant time effect found in HR/Pslope, Ppeak, Pmax, and HRstmax; the corresponding statistical data are presented in Table 2.

Pmax and HRstmax exhibited a clear and significant biphasic response, being largely decreased on day0 and moderately increased on day1 compared to baseline; no differences were found later after PE (Table 2). HR/Pslope and Ppeak decreased significantly on day0 compared to baseline and did not differ compared to baseline at any other time points (Table 2).

### 3.5. Subjective Markers

There was a significant time effect found in RPE and VAS; the corresponding statistical data are presented in Table 2. The subjects reported a moderate increase in RPE on day0 compared to baseline; however, a moderate decrease in RPE was detected on day7. Leg soreness (VAS) was significantly increased on day0 and day1 compared to baseline (Table 2).

### 3.6. Other Parameters

We found no significant time effect neither in TBW nor in ECW (Table 2).

BFs of each participant pre- and post-SSE over all five measuring points were confirmed to fall inside the HF band of frequency domain HRV (Table 2).

### 3.7. Relationships between Cardiac Function Indices and Exercise Performance

There were no clear correlations between HR and any of the selected parasympathetic activity indices (lnRMSSD preSSE, lnRMSSD postSSE, lnHF preSSE, HRR30, HRR60) when examining each SSE over the trial separately. Even when all data pooled after PE were combined, absolute values of cardiac parasympathetic activity indices listed above did not correlate either with exercise performance (Pmax, HRstmax) or with RPF (Table 3). The only parameter exhibiting correlations with aerobic performance indices and RPE after pooling was HR. It correlates moderately to HRstmax either measured before SSE or in the recovery after SSE. A small correlation exists between RPE and both, HR preSSE and postSSE. HRpreSSE but not HRRpostSSE correlate with predicted peak power output (Pmax), and the correlation was found to be small and negative (Table 3).

A small negative correlation was observed between Pmax and HRstmax, and a moderate correlation was found between Pmax and RPE, respectively (Table 3).

### 3.8. Relationships between the Changes of Cardiac Function Indices and Exercise Performance

No clear correlations between any of the selected parameters’ changes were found when examining each SSE over the trial separately. Pooling all data obtained after PE, the following correlations between individual changes were observed: The change in Pmax correlated largely to the change in HR preSSE, moderately to the changes in lnRMSSD pre- and postSSE, HR postSSE, as well as to HRR60. Small correlations of the changes in lnHF preSSE and HRR30 with Pmax% were addressed. Pmax% did not correlate with lnHF% postSSE (Table 4);The change in HRstmax correlated with a very large r value to the change in HR preSSE, large to the changes in lnRMSSD preSSE and HR postSSE, moderate to lnRMSSD% postSSE, HRR30%, and HRR60% and small to lnHF% preSSE. HRstmax% did not correlate with lnHF% postSSE (Table 4);The change in RPE correlated small and negative to lnRMSSD% preSSE and HRR30%. No correlations between RPE% and other cardiac parasympathetic activity-related indices were determined. (Table 4).

### 3.9. Multiple Indices Assessment

All together, we have demonstrated that PE leads to a statistically significant biphasic kinetics in HRpostSSE, HRR30, HRR60, Pmax, and HRstmax regarding day0 and day1 response to SSE with respect to baseline (Figure 2 and Figure 4, Table 4). Additionally, lnRMSSD postSSE exhibited a biphasic response; however, this did not reach statistical significance. The changes in cardiac and performance indices with biphasic kinetics following PE with respect to baseline are presented in hexagonal plots (Figure 4) together with lnRMSSD pre- and postSSE, albeit their bivariate change was not statistically significant. The change in lnHFpreSSE was included in the hexagonal plot as it is often promoted as a predictor of athletes’ training status [32].

## 4. Discussion

In the present study, we investigated the reorganization of cardiac autonomic adjustment to short-lasting submaximal exercise bouts following a prolonged exertion (half-marathon run). This study aimed to contribute to the understanding of short-term cardiac parasympathetic control regarding the training history. We have found that training history importantly affects the recovery after short-lasting submaximal exercise. Further, the impact of prolonged exertion as a part of training history on subsequent cardiac autonomic dynamics was augmented in response to short-lasting submaximal exercise compared to rest.

The primary findings are as follows:Cardiac parasympathetic reactivation following short-lasting submaximal exercise subsequent to a half-marathon run exhibited biphasic response on a daily basis, being suppressed when SSE was performed on the same day and overexpressed on the first day after the half-marathon.We observed moderate biphasic changes in associated aerobic performance on a daily basis, which is potentially important for elite athletes.HRR as an index of parasympathetic reactivation depends on training history, not only on exercise intensity.HRR and HR in recovery after short-lasting submaximal exercise reflected training history better than resting values.We observed a correlation between the changes in resting and post-short-term submaximal exercise parasympathetic indices throughout recovery after a half-marathon with respect to baseline values and corresponding changes in aerobic power output indices may predict aerobic performance at least one week after prolonged exertion (heavy training) in recreational runners.Considering the change of multiple parameters reflecting current cardiac autonomic activity and an athlete’s performance with respect to training history (postSSE HR, HRR, lnRMSSD before and after short-lasting submaximal exercise, resting lnHF and aerobic capacity) offers a more detailed insight into training recovery.

Immediate and prolonged effects of exhaustive exercise on cardiovascular autonomic modulation have been demonstrated previously by many authors [2,14,33,34]. Cardiac autonomic readjustment to the short-lasting submaximal exercise repeated successively after exhaustive exercise, however, has not previously been employed in a prolonged observational period. Our results showed time dependent cANA kinetics: autonomic dysfunction characterized by impairment of cardiac parasympathetic activity on the same day followed by parasympathetic overactivation to the heart one day after a half-marathon. HR post-SSE and HRR were found to reflect exercising history better than any other cANA indices in line with the findings of Buchheit and his colleague, who proposed HRR as a more sensitive marker of recently applied training loads than other cPS indices. Additionally, associated exercise performance indices (Pmax and HRstmax) exhibited the same day-by-day pattern throughout one week follow-up after exhaustive exercise. Parasympathetic impairment on the same day was associated with attenuated power output and increased Hrstmax yet parasympathetic overactivation one day after PE with augmented power output and decreased Hrstmax, respectively.

### 4.1. Same Day Response

In our study, higher HR, lower lnRMSSD, and lnHF together with a clear shift towards a sympathetic predominance (lnLF/HF) characterized the resting state one hour after completing a half-marathon compared to baseline, indicating that cardiac parasympathetic activity is attenuated. A sympathetic predominance throughout this period is consistent with the findings of Vecchia and colleagues [35], who proposed that sympathetic predominance outlasts the period of exercise. These findings correspond to the reduced baroreflex sensitivity observed 1 h after a half-marathon in healthy subjects [36]. A possible explanation for the same day increased cardiac sympathetic activity could be attributed to high levels of metabolites, especially lactate [4], and increased body temperature due to physical exercise after half-marathon run cessation [37], which both might be at least in part responsible for the delay in the cardiac autonomic recovery keeping metaboreflex and thermoregulation activated even after exercise cessation [4].

Thus, submaximal exercise performed on the same day required further adaptation of cANA superimposed upon initially depressed parasympathetic and potentially elevated sympathetic drive and is characterized by slower HRR, increased HR, and decreased lnRMSSD postSSE, yet did not lead to additional changes in either lnHF or lnLF/HF after SSE cessation compared to baseline and to preSSE, accordingly.

These results potentially indicated short-term cardiac autonomic adaptability impairment, attributed to cardiac sympathetic exhaustion or linked to the additional delay in sympathetic inhibition caused by prolonged exertion [38]. Such cardiac sympathetic adaptability impairment was also reported by Hara et al. in response to symptom-limited exercise in heart failure, where increased sympathetic activation after such exercise was not accompanied by the correspondent change in hemodynamic indices [38].

As evident from reduced HRR on day0, vagal dysfunction occurred simultaneously and may be interpreted as a decreased ability to recruit parasympathetic cardiac vagal tone when sympathetic drive to the heart is elevated and/or when parasympathetic drive to the heart is too low, respectively [17].

However, because HRV reflects only the end response that integrates different autonomic facilitatory and inhibitory effects, the underlying mechanisms remain speculative.

According to some large clinical studies that demonstrated a strong association between vagal dysfunction, cardiovascular morbidity, and all-cause mortality [39], this early period after PE could be referred as a vulnerable period. Additionally, a marked day0 decrease in the difference between pre- and postSSE HR compared to baseline and to all following SSE bouts, was strongly and independently associated with sudden cardiac death [39].

Thus, a practical application to training strategies is issued, namely, to avoid any further exercise on the same day after prolonged exhaustive training or, if inevitably due to the competition schedule, to apply techniques for accelerated recovery such as cryotherapy or cold water immersion to minimize the vulnerability of this particular state as proposed by Van Hooren [11]. Further, high-frequency training programs are recommended to improve HRV in older adults [40] characterized by age-related decline in HRV; however, based on the results of our study, the training frequency should not exceed one training session per day or should be even less frequent, given that the vulnerable period could potentially be prolonged in older adults due to less reactive ANS. Further studies are needed to prove this speculation.

Concerning exercise performance, day0 SSE is accompanied by largely reduced aerobic power output compared to baseline. These results corroborate the common opinion that sympathetic overactivation is associated with poor performance in connection with overreaching and overtraining [8] or in chronic heart failure patients [39,41,42,43]. The precise mechanism underlying decreased performance at higher sympathetic outflow is not completely clear [44]; nevertheless, it is obvious that adaptation to exercise upon sympathetic overactivation at its onset is compromised [12] and that interplay of simultaneous sympathetic overactivation and parasympathetic decline may have a role. On the other hand, increased sympathetic tone positively correlates with the competitive performance [45] substantiated by better coordination, increased cardiac output, blood flow redistribution, and perfusion of exercising muscles [46,47].

In our study, RPE and leg muscle soreness were increased on the same day after the half-marathon. This correlates well with the results of Wiewelhove and colleagues, who demonstrated that higher overall stress and VAS after active recovery following a half-marathon were related to alterations in muscle contractile characteristics, blood markers of muscle damage (creatine kinase), inflammation (C-reactive protein), and metabolic status (insulin like growth factor 1) as well as to altered perception of muscle soreness [6].

### 4.2. One Day after Half-Marathon

One day after the half-marathon, an accentuated rebound phenomenon was observed by increased HRR, reflecting enhanced vagal reactivation compared to the pre-exercise level. This is the original finding of the present study.

The HRR increase was accompanied by the decreased HR later in the recovery after SSE, but not accompanied by statistically significant changes in any other cPS related HRV parameters, neither in time nor in frequency domain at rest before SSE. Accordingly, day1 preSSE HR did not differ statistically from the baseline value. These results lead to the conclusion that cPS activity is completely recovered 24 h after the half-marathon but its ability to readapt to another provocation is affected by exercise history. Consequently, biphasic changes in HRR and HR in response to SSE after PE indicate their potential sensitivity in assessing the cumulative exercise stress compared to resting HR and HR-related indices.

To date, an abundance of evidence demonstrated increased cPS-related HR indices at rest following prolonged exertion; however, their pre-exercise values were not exceeded [9] unless recovery acceleration strategies were applied. The increase in preSSE cPS HRV derived indices (lnRMSSD an lnHF) over baseline values (Figure 3) observed in our study on day 1 and day 2 after PE advocate for the use of SSE as an efficient active cool-down strategy.

Similar increases in cPS-derived HRV indices at rest following prolonged exertion were demonstrated in response to high-endurance or supramaximal exercise [5,48]. Hautala and colleagues demonstrated that HF values 48 h after 75 km cross-country skiing exceeded the pre-race level in young healthy adults [5]. Additionally, rebound of post-exercise cardiac parasympathetic activity above pre-exercise levels has been observed two days after a supramaximal exercise session in moderately trained males [48]. The delay of rebound onset is possibly attributed to the different exhaustive exercise pattern employed in these studies.

The precise mechanism behind this rebound effect of cPS activity is still not clear, but is most likely dependent on exercise-induced changes in plasma volume and resultant arterio-baroreflex stimulation [9]. However, in our study, no increase in extracellular or total water volume was detected indicating that some other mechanisms must be involved.

Exercise-induced parasympathetic overcompensation could be related either to positive or negative cardiorespiratory adaptations [12]. Kiviniemi considered this parasympathetic rebound period after prolonged exertion as the optimal training period for attaining cardiorespiratory adaptations; however, it may also lead to overreaching [32]. Based on these findings, many studies provided evidence in support of the usefulness of HRV recording during chronic exercise to optimize training adaptations and enhance gain in performance [12,49]. According to the results of our study, in line with Peçcanha et al. [16] and Ballinger [8], vagal reactivation, reflecting in HRR, and vagal modulation, reflected in cPS cardiac indices and HR after exercise, could potentially reveal the difference between both possible scenarios; however, performance and fatigue are still proposed as major parameters distinguishing between both states [8], confirming the complexity of physiological mechanisms involved.

Our study showed that on day1 after prolonged exertion, positive cardiorespiratory adaptations occurred, confirmed by moderate gain in the performance indices, Pmax, and HRstmax without any increase in fatigue. This finding of our study could be possibly transferred to practice by encouraging recreational runners to exercise 24 h after strenuous exercise to improve their cardiorespiratory fitness and potentially gain improvements in performance.

Regarding general fatigue and muscle soreness on day1 following the half-marathon, RPE is decreased toward baseline value whereas VAS remained increased. These results are consistent with the findings of Wiewelhove and co-authors [6], who demonstrated that increased muscle soreness onday1 after a half-marathon is related to increased creatine kinase in blood as a marker of muscle damage. Interestingly, it was found in the same study that active cool-down after a half-marathon does not improve recovery in this aspect; hence, less muscle damage was found in passive recovery and other non-native recovery strategies compared to active cool-down.

### 4.3. Day2 and 1 Week after Half-Marathon

Almost all measured parameters pre- and postSSE retain their baseline values 48 h following the half-marathon. The only exception is postSSE HR on day2 being still significantly decreased compared to baseline. An elevation and drop in HR post SSE after termination of exercise on day0 and day1, respectively, could be a plausible consequence of vagal depression followed by its reactivation; however, on day2, all HRV-derived indices as well as HRR are recovered. Hypothetically, this could be attributed to sympathetic withdrawal on day 2 after strenuous exercise, yet this is not confirmed with our results.

### 4.4. Relationships between Cardiac Function Indices and Their Changes and Exercise Performance

The relationship between cardiac function indices (HR, HRV, and HRR) and exercise performance is not fully understood. We have shown that short-term cANA dynamics are reflected in the changes of aerobic performance on a daily basis apart from long-term changes adopted by exhaustive training programs lasting several weeks. As mentioned earlier, aerobic exercise performance is highly influenced by the sympathetic drive, yet sympathetic overactivation leads to poorer performance [8]. On the other hand, augmented resting cardiac parasympathetic tone obtained by regular endurance training increases performance by long-term cardiac adaptations such as increased left ventricular internal dimensions, wall thickness, as well as by noncardiac gain in muscle capillarisation, increased plasma volume, and decreased peripheral resistance, resulting in decreased metabolic load during exercise. However, parasympathetic hyperactivity may limit the ability to fully engage the sympathetic nervous system during maximal exercise [12] like in parasympathetic overtraining. We have not found any significant correlations between cardiac function indices and performance when examining each particular SSE over the whole experimental trial separately, probably due to a small sample size. When the data of all SSE bouts during the experimental trial were pooled together, exercise performance indices did not correlate with any of the cardiac parasympathetic parameters. We can speculate that pooling the data blunted the partial effects of different cANA responses to SSE before and after PE over the experimental trial. Additionally, participants of our study were of different age, gender, and fitness level implicating their high individual variability of HRV indices including HRR. These findings are in conflict with the conclusions of Buchheit and his colleague who found that cPS-related HRV indices are related to cardio-respiratory fitness [18]. This conflict may be attributable to the fact that HRV in their study was measured at rest after abstaining from exercise for 2 days in all participants. The same study reported no correlation between HRR and cardio-respiratory fitness, corroborating with our results.

The only index of cardiac function exhibiting the correlation with exercise performance, though small and negative (Table 3), was resting HR, which reflects the coordinated interaction between cardiac parasympathetic and sympathetic activity in a resting state. This is in compliance with the mainly accepted opinion that resting HR represents a robust surrogate of aerobic fitness level [50].

We found small to moderate correlations between the change in all pre- and postSSE cardiac pSA indices except lnHF postEES (Table 4). A correlation of this magnitude requires careful interpretation. However, this association suggests that the greater the increase in either cPS modulation or reactivation, the greater the increase in Pmax and the decrease in HRstmax. These findings are in line with the conclusions of many authors [12] who found increased exercise performance associated with increased cPS related indices yet attributable to long term cPS adaptations built up during several weeks of endurance training. However, there are plenty of opposing reports, where the increase in cPS activity was found to provoke a decrease in aerobic power output [12] associated with overreaching or overtraining. Based on these results, we may conclude that regarding the gain in performance, short-term cPS adaptations following prolonged exertion observed in our study mimic long-term adaptations with improved performance.

Further, the changes in HR pre- and postSSE show a large to moderate correlation with Pmax and a very large to large correlation with HRstmax, confirming HR as a robust surrogate to determine the change of power output.

We have found that irrespective of pre-exercise cPS activity, the greater the increase in HRR30 and lnRMSSD preSSE, the greater the reduction in RPE, yet the correlations were small.

## 5. Conclusions

We have measured the post-exercise heart rate and HRV changes in response to short-lasting submaximal exercise bouts repeated throughout one-week after a half-marathon. The half-marathon served as a model of prolonged exertion, which potentially affects the autonomic response to subsequent exercise for days. Our main finding was that cardiac autonomic response to submaximal exercise differed with respect to training history: autonomic imbalance with depressed HRV and HRR was found on day 0 after prolonged exertion, whereas 24 h after it, parasympathetic rebound occurred. Interestingly, the associated power output changed simultaneously. It was shown that the recovery after submaximal exercise is more sensitive to exercise history than resting cardiac autonomic activity.

The findings of our study could be possibly transferred to other pathophysiological states characterized by cardiac autonomic dysfunction, such as chronic heart failure (CHF) and overreaching or overtraining in athletes. CHF is associated with depressed HRV compared to healthy state at rest [51], decreased HRR, and performance [52]. Based on our results, it could be speculated that PE provokes CHF like cardiac autonomic modulation and could serve as a studying model for CHF. The loss of parasympathetic reactivation after exercise cessation is associated with increased incidence of arrhythmic deaths [51], so an increased risk for sudden adverse cardiac events could be predicted when exercising early after PE.

The day0 cardiac autonomic response to submaximal exercise copes with the cardiac manifestation of sympathetic overtraining syndrome [53]. A lot of studies confirmed that parameters of cardiac parasympathetic activity, such as RMSSD, HF, or their natural logarithms and HR at rest [4,8,12] are indicators of the performance improvement or regression yet are not univocal. Based on the outcomes of our study, it could be concluded that RMSSD and HR measured after submaximal dynamic exercise are more powerful in predicting performance than resting values.

The results of our study are potentially applicable for athletes in planning their daily training activities and scheduling heart rate monitoring to potentially determine the time window for efficient cardiovascular performance improvement as well as to recognize the deviated cardiac autonomic patterns, which could potentially indicate cardiac vulnerability or early signs of overtraining.

## Figures and Tables

**Figure 1 ijerph-19-09797-f001:**
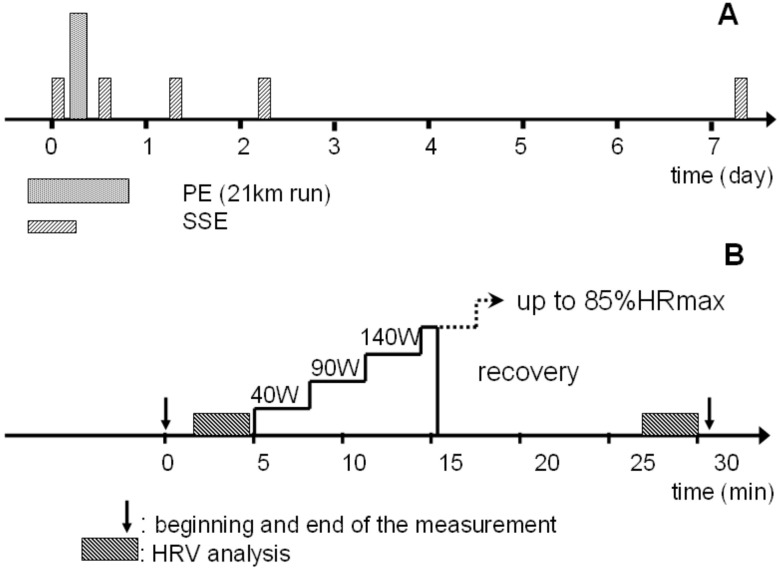
Overview of experimental trial protocol; (**A**) timeline of exercises performed by subjects; PE—prolonged exertion in the form of a half-marathon run; SSE—short-lasting submaximal exercise; (**B**) timeline of short-lasting submaximal cycling exercise; HRmax—maximal heart rate; HRV—heart rate variability.

**Figure 2 ijerph-19-09797-f002:**
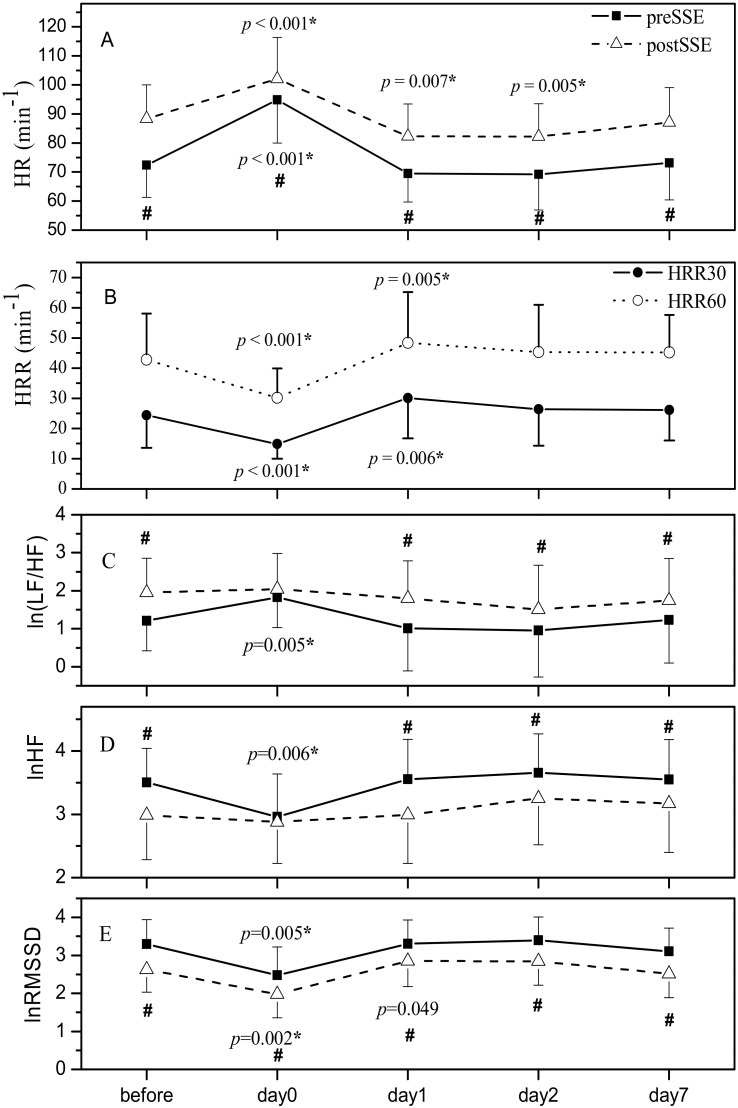
The time course of HR (**A**), HRR (**B**), ln(LF/HF) (**C**), lnHF (**D**), and lnRMSSD (**E**) measured preSSE and postSSE over all five measuring timepoints during experimental trial (before—before PE (baseline), day0—1 h after PE, day1—1 day after PE, day2—2 days after PE, day7—one week after PE). HR—heart rate; HRR30—heart rate recovery in 30 s; HRR60—heart rate recovery in 60 s; ln(LF/HF)—ln transform of the ratio of low-frequency power to high-frequency power in frequency domain heart rate variability; lnHF—ln transform of high frequency power of frequency domaine heart rate variability measured in normalized units; lnRMSSD—ln transform of root mean square of successive interval differences of time domaine heart rate variability; PE—prolonged exertion; SSE—short-lasting submaximal exercise. Values are represented as mean ± standard deviation. *—statistically significant compared to baseline (before), *p*—corresponding *p* value; #—statistically significant difference postSSE versus preSSE.

**Figure 3 ijerph-19-09797-f003:**
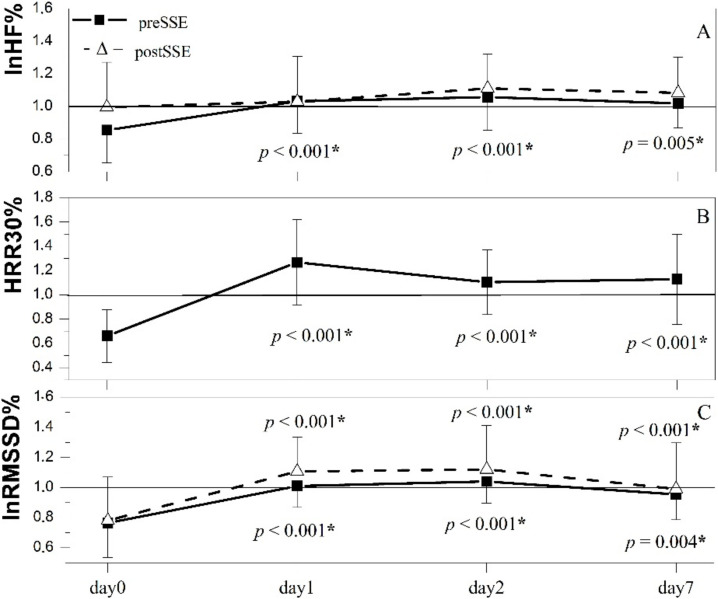
The time course of the change in cardiac parasympathetic activity indices versus baseline represented by lnHF% (**A**), HRR30% (**B**), and lnRMSSD% (**C**) over the experimental trail (day0—1 h after PE, day1—1 day after PE, day2—2 days after PE, day7—one week after PE). lnHF%—the ratio of ln transform of high-frequency power of frequency domain heart rate variability measured in normalized units at each particular time point after PE and corresponding baseline value; HRR30%—the ratio of heart rate recovery in 30 s after exercise cessation at each particular time point after PE and corresponding baseline value; lnRMSSD%—the ratio of ln transform of root mean square of successive interval differences of time domain heart rate variability at each particular time point after PE and corresponding baseline value; PE—prolonged exertion; SSE—short-lasting submaximal exercise. Values are represented as mean ± standard deviation. *—statistically significant compared to day0, *p*—corresponding *p* value.

**Figure 4 ijerph-19-09797-f004:**
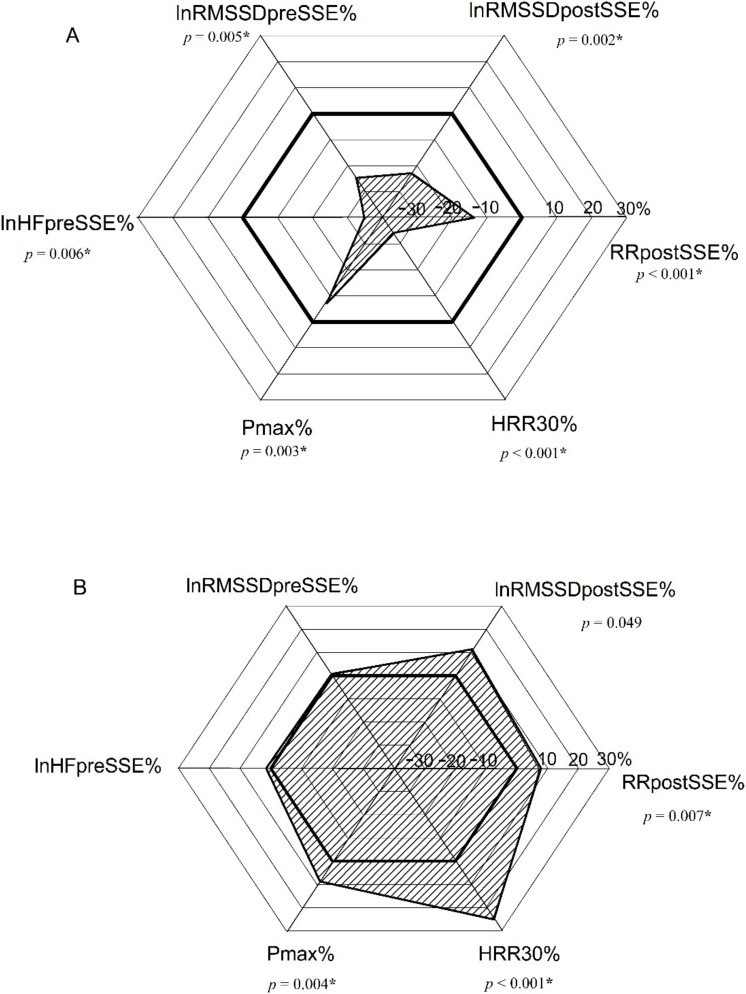
The change in mean value of six selected indices on the same day (**A**) and one day following PE (**B**) with respect to baseline, represented in hexagon graph. Pmax%—the ratio of maximal power output at each particular time point after PE and its baseline value; HRR30%—the ratio of heart rate recovery in 30 s at each particular time point after PE and its baseline value; lnRMSSD%—the ratio of ln transform of root mean square of successive interval differences of time domain heart rate variability at each particular time point after PE and its baseline value; lnHF%—the ratio of ln transform of high frequency power of frequency domain heart rate variability measured in normalized units at each particular time point after PE and its baseline value; RR%—the ratio of cardiac cycle duration (HR^-1^) at each particular time point after PE and its baseline value; PE—prolonged exertion; SSE—short-lasting submaximal exercise; *p*—corresponding *p* value; *—*p* < 0.05. Bold line represents baseline values.

**Table 1 ijerph-19-09797-t001:** Baseline characteristics of recreational runners participating in the study (n = 19).

Variables	Values
Age (years)	40.4 ± 15.2
Weight (kg)	72.7 ± 15.3
Height (cm)	176.8 ± 10.6
BMI (kg/m^2^)	23.0 ± 2.7
Physical activity of any kind (days per week)	4.3 ± 1.2
Running training (days per week)	2.5 ± 1.2
Mean distance per running training (km/session)	11.2 ± 3.8
21 km run score (min)	126.6 ± 19.1
Mean HR during 21 km run (%HRmax)	72 ± 11

Values are represented as mean ± standard deviation.

**Table 2 ijerph-19-09797-t002:** Subjective recovery, indices of aerobic performance, hydration, and breathing frequency over time.

	Before	Day0	Day1	Day2	Day7	F	*p*	η^2^
**RPE**	13.17 ± 2.09	14.11 ± 2.88	13.94 ± 2.88	13.06 ± 2.94	12.50 ± 2.57	(2.74; 13.88) 5.95	<0.001	0.25
**ES**		0.52; *p* < 0.001 moderate			0.51; *p* < 0.001 moderate			
**VAS**	2.63 (1.50;3.76)	16.05 (7.86;24.26) * *p* < 0.001	13.26 (7.44;19.08) * *p* = 0.006	11.63 (5.13;18.13)	3.05 (1.13;4.97)		<0.001	
**HR/P slope** **(min^−1^/W)**	0.42 ± 0.10	0.36 ± 0.09 *	0.39 ± 0.08	0.43 ± 0.12	0.42 ± 0.11	(4;68) 11.77	<0.001	0.41
**ES**		1.147 *p* < 0.001 large						
**Ppeak** **(W)**	208 (184;233)	184 (159;210) * *p* = 0.010	221 (196;247)	211 (187;236)	205 (178;234)		0.002	
**Pmax** **(W)**	269 ± 74	249 ± 67 *	296 ± 63 *	279 ± 70	271 ± 69	(4;68) 10.21	<0.001	0.38
**ES**		0.81; *p* = 0.003 large	0.78; *p* = 0.004 moderate					
**HRstmax (min^−1^)**	134.2 ± 14.7	146.4 ± 10.8 *	130.1 ± 13.8 *	134.1 ± 15.0	132.4 ± 16.6	(4;72) 28.205	<0.001	0.59
**ES**		1.33; *p* < 0.001 large	0.74; *p* = 0.005 moderate					
**TBW (L)**	42.7 ± 10.7	42.66 ± 10.7	43.33 ± 11.3	42.53 ± 10.9	42.09 ± 10.8	(2.26;36.26) 1.84	0.17	0.10
**ECW (L)**	18.08 ± 4.40	18.13 ± 4.18	17.46 ± 4.41	18.29 ± 4.47	17.28 ± 3.99	(1.92;30.78) 0.79	0.46	0.05
**BF (min^−1^) preSSE**	14.23 ± 3.83	17.61 ± 4.65 *	15.28 ± 4.17	14.99 ± 4.00	14.51 ± 3.39	(4;72) 9.89	<0.001	0.36
**BF (min^−1^)** **postSSE**	16.55 ± 3.58	19.18 ± 4.25 *	17.03 ± 4.18	16.58 ± 4.02	16.35 ± 3.71	(4;72) 10.69	<0.001	0.37

Before—before PE (baseline); day0—1 h after PE; day1—1 day after PE; day2—2 days after PE, day7—one week after PE; PE—prolonged exertion; RPE—rate of perceived exertion; VAS—leg soreness on a visual analogue scale; HR/Pslope—steady state heart rate versus power output linear relationship coefficient; Ppeak—peak power output achieved; Pmax—maximal aerobic power output; HRstmax—maximal steady state heart rate; TBW—total body water; ECW—extracellular water volume; BF—breathing frequency; ES—effect size; F, *p* and η^2^—parameters of rANOVA. *—statistically significant. Values are represented as mean ± standard deviation when normally distributed and as mean (lower bound of 95% confidence interval of means; upper bound of 95% confidence interval of means) when not distributed normally.

**Table 3 ijerph-19-09797-t003:** Relationship between cardiac parasympathetic activity indices, heart rate, performance, and RPE combined for all data over the experimental trial.

		Pmax		HRstmax		RPE	
	n	p	r	p	r	p	
**HRR30**	95	0.739		0.089		0.963	
**HRR60**	95	0.253		0.815		0.499	
**lnRMSSD preSSE**	95	0.950		0.473		0.114	
**lnRMSSD postSSE**	95	0.527		0.672		0.137	
**lnHF preSSE**	95	0.117		0.099		0.089	
**HR preSSE**	95	0.007	−0.283 small	<0.001 *	0.458 moderate	0.037 *	−0.214 small
**HR postSSE**	95	0.290		<0.001 *	0.492 moderate	0.039 *	−0.212 small
**HRstmax**	95	0.022 *	−0.241 small			0.454	
**RPE**	95	0.003 *	0.308 moderate	0.454			

Pmax—maximal power output; HRstmax—maximal steady state heart rate; RPE—rate of perceived exertion; HRR30(60)—heart rate recovery in 30 s (60 s); lnRMSSD—ln transform of root mean square of successive differences of time domain heart rate variability; lnHF—ln transform of high frequency power of frequency domain heart rate variability measured in normalized units; HR—heart rate; SSE—short-lasting submaximal exercise. n—number of points; r—Pearson’s coefficient and the interpretation of the correlation magnitude; p—corresponding *p* value; *—*p* < 0.05.

**Table 4 ijerph-19-09797-t004:** Relationship between the change in all measured cardiac indices, the change in performance, and in RPE with respect to baseline values combined for all data after prolonged exertion.

		Pmax%		HRstmax%		RPE%	
	n	p	r	p	r	p	r
**HRR30%**	76	0.049 *	0.225 small	<0.001 *	−0.431 moderate	0.022 *	−0.263 small
**HRR60%**	76	0.001 *	0.374 moderate	0.006 *	−0.311 moderate	0.854	
**lnRMSSD% preSSE**	76	<0.001 *	0.470 moderate	<0.001 *	−0.504 large	0.0433 *	−0.234 small
**lnRMSSD% postSSE**	76	0.003 *	0.343 moderate	0.001 *	−0.364 moderate	0.071	
**lnHF% preSSE**	76	0.031 *	0.254 small	0.012 *	−0.287 small	0.729	
**lnHF% postSSE**	76	0.434		0.168		0.344	
**HR% preSSE**	76	<0.001 *	−0.522 large	<0.001 *	0.749 very large	0.204	
**HR% postSSE**	76	<0.001 *	−0.487 moderate	<0.001 *	0.591 large	0.211	
**HRstmax%**	76	<0.001 *	−0.490 moderate			0.910	
**RPE%**	76	0.933		0.910			

Pmax%—the ratio of maximal aerobic power output at each time point after PE and its baseline value; HRstmax%—the ratio of maximal steady state heart rate at each time point after PE and its baseline value; RPE%—the ratio of rate of perceived exertion (RPE) at each time point after PE and its baseline value; HRR30(60)%—the ratio of heart rate recovery in 30(60) s at each particular time point after PE and its baseline value; lnRMSSD%—the ratio of ln transform of root mean square of successive interval differences of time domain heart rate variability at each particular time point after PE and its baseline value; lnHF%—the ratio of ln transform of high frequency power of frequency domain heart rate variability measured in normalized units at each particular time point after PE and its baseline value; HR%—the ratio of heart rate at each particular time point after PE and its baseline value; SSE—short-lasting submaximal exercise; PE—prolonged exertion; n—number of points; r—Pearson’s coefficient and the interpretation of the correlation magnitude; p—corresponding *p* value; *—*p* < 0.05.

## Data Availability

The data that support the findings of this study are available from the corresponding author upon reasonable request.

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
