# Peer review of "Training History, Cardiac Autonomic Recovery from Submaximal Exercise and Associated Performance in Recreational Runners"

_ijerph, 2022, doi:10.3390/ijerph19169797_

Round 1

Reviewer 1 Report

The main aim of the paper „Training history, cardiac autonomic recovery from submaximal exercise and associated performance in recreational runners “ was  focused on how different pre-exercise cANA provoked by exhaustive training influences cardiac autonomic response to successive submaximal dynamic exercise sessions, associated power output and RPE. The authors also aimed to test if any correlations exist between the changes in pre/post-exercise HR derived indices and performance to potentially advocate for the short-lasting submaximal exercise bouts to be included in regular training regime for monitoring athlete’s performance.

The study is interesting. I would like to appretiate the efforts of the authors. However, some information is missing in the article, some facts need to be explained:

Major comments:

Some methodologies need to be explained:

Line 135: What was the intensity of the 21.1 km run? Did they run with maximum effort? Or how fast? Was the speed somehow determined?

Line 149-150: How was HRmax defined or measured, of which 85% was calculated?

Minor comments:

The abstract: The abbreviation HRV is not explained.

Line 25: … small to very large correlations …  - I recommend explaining or clarifying.

Line 106-115: I recommend to better specify the first sentence of the goal of the work.

Line 123: Would not it be more correct to give BMI values for men and women separately?

Conclusion: I recommend to better formulate the conclusions, include specific knowledge.

References: There are other studies in the last 5 years dealing with similar topics. I recommend some topical ones to include.

Reviewer 2 Report

Training history, cardiac autonomic recovery from submaximal exercise and associated performance in recreational runners.

The authors present an interesting study measuring post exercise heart rate and HRV changes in response to short-lasting submaximal exercise bouts repeated throughout a week after a half marathon. There are some interesting and important findings that the authors report on. The most interesting note is that the authors demonstrate the rebound phenomenon where the athletes saw an increase in peak power output 24 hours following the half marathon. This could be used by athletes to understand and use this time to reach their full potential. Most parameters measured were returned to baseline by 48 hours, with only postSSE GR at day 2 still significantly decreased, which could indicate vagal depression followed by reactivation and sympathetic withdrawal after strenuous exercise, but this requires further exploration.

This could certainly be useful for athletes who are looking to gain fitness and also understanding over-reaching and overtraining.

There are however, a number of question and comments listed below for the authors to address.  

Abstract line 16. With cardiac indices was tested. Cardiac indices is plural and therefore was should be changed to were. The commas in the numbers on line 22 don’t make sense, should these be .?

Line 71: reword from however, no one tracked – to there is little no data on the short term….

Could the authors explain why they used cycling in the week following the half marathon.

Line 122 and 123 please use units with measures of BMI. In line 127, please use units for HR and remove, and use .

Methods

Cross over study? It seems that this is not a cross over study. A cross over study is where there are two arms of the study, and the participant completed both arms one as a control and the other as an experimental and there is a washout period in between. This was not the case with this study.

20 recreational runners training more than 3 times a week. What other characteristics were important for this group of recreational runners? How long did they need to run for each week? Did they need to cover a certain amount of km, or have a certain speed? It would be pertinent for the reader to be provided with more details about the amount and duration of exercise that individuals were currently engaged in prior to the experiment. How long had the individuals been needing to be active for?

Why did the researchers decide not to use the Karvonen formula for calculating maximal heart rate for everyone?

First exercise test – Running a half marathon (21km) on a predefined track outside.

Graded exercise testing

Short lasting submaximal exercise bouts -  was completed before, one hour, 24 hours, 24 hours and 8 days after.

Each session consisted on 5 minute of sitting in a rest position, followed by graded exercise test on cycle ergometer – 40W for 3 minutes, increase of 50W every 3 minutes until target HR peak was achieved (85% HR max) – seated position for 15 minutes afterwards.

Space needed on line 144 in between 24 and hours… and then again on line 46 in between 5 and minutes.

HRV

RR interval taken from the last 3 minutes of sitting and 12th – 15th minute following cycling. LF/HF ratio

HR recovery in 30 and 60s were defined as difference between peak HR and HR recorded at these time points following exercise.

Indices of aerobic performance – maximal power output – determined form HR vs power relationship in steady state.

Borg scale and visual analog scale

Bioelectrical impedance

The article reports that the study is a cross over design, but it does not state how this was completed. Is my understanding wrong in the timeline I have below?

All subjects completed a graded exercise test to 85% max.

One hour later completed the half marathon.

Graded exercise test completed one hour after half marathon

Graded exercise test completed 24 hours after half marathon

Graded exercise test completed 48 hours after half marathon

Graded exercise test completed 8 days after half marathon

HRV and HR were monitored consistently but HRV timepoints minutes 3-5 during pre-exercise data and then minutes 12-15 post graded exercise test at each of the timepoints above. The Borg scale and visual analog scale were used during each graded exercise test.

Bioelectrical impedance was performed before each graded exercise test.

In table 1, please use . instead of , in between values. Ensure that BMI has units attached to it (kg.m2). In Table 1, it states mean distance per running training in km, is this per week or per session/day? Please clarify this.

The study demonstrates following a half marathon that it takes approximately 7 days to recover in Hr and HRR. In LF/HF this is still elevated at 7 days, but reduced pre exercise. HF and RMSSD drop significantly following the half marathon, but are not significantly different at day 1-7.  HR is significantly different at 24 and 48 hours, but not at 7 days.

In the aerobic indicators, peak power notably decreased one hour post exercise. Interestingly on day 2 peak power output was at its highest and then decrease at day 2 and back to usual on day 7. Therefore, evidencing rebound phenomenon.

Prolonged exertion -

Complexity of physiological mechanism involved in performance and fatigue.

It is interesting that most of the parameters measured returned to their baseline values 48 hours following half marathon.

Round 2

Reviewer 1 Report

The authors of the article answered the questions and incorporated the comments, but there are still a few facts that need to be added or explained:

- Information about the determined intensity of running must be given in the methodological part of the work. The explanation in the cover letter is sufficient, but this information is also important in the text of the article. Information about the realized speed of running cannot replace it, because it says nothing about the subjective effort of running.

- Theoretically determined HRmax is usable, but in practice highly inaccurate. However, if you are based on published conclusions (see cover letter), it is necessary to citate in the text.
